# How Can Ice Emerge at 0 °C?

**DOI:** 10.3390/biom12070981

**Published:** 2022-07-13

**Authors:** Alexei V. Finkelstein, Sergiy O. Garbuzynskiy, Bogdan S. Melnik

**Affiliations:** 1Institute of Protein Research, Russian Academy of Sciences, 142290 Pushchino, Russia; sergey@phys.protres.ru (S.O.G.); bmelnik@phys.protres.ru (B.S.M.); 2Faculty of Biotechnology, Lomonosov Moscow State University, 142290 Pushchino, Russia; 3Faculty of Biology, Lomonosov Moscow State University, 119192 Moscow, Russia

**Keywords:** freezing point of water, time of freezing, melting point of ice, ice nucleation, *Pseudomonas syringae*, ice-binding protein

## Abstract

The classical nucleation theory shows that bulk water freezing does not occur at temperatures above ≈ −30 °C, and that at higher temperatures ice nucleation requires the presence of some ice-binding surfaces. The temperature and rate of ice nucleation depend on the size and level of complementarity between the atomic structure of these surfaces and various H-bond-rich/depleted crystal planes. In our experiments, the ice nucleation temperature was within a range from −8 °C to −15 °C for buffer and water in plastic test tubes. Upon the addition of ice-initiating substances (i.e., conventional AgI or CuO investigated here), ice appeared in a range from −3 °C to −7 °C, and in the presence of the ice-nucleating bacterium *Pseudomonas syringae* from −1 °C to −2 °C. The addition of an antifreeze protein inhibited the action of the tested ice-initiating agents.

## 1. Introduction

On our planet, liquid water is an extremely widespread compound. It is crucially important for all organisms living here. Thus, a transformation of liquid water into ice has serious consequences for the affected organisms, and since conditions that favor water-to-ice transition exist (at least sometimes) on a large fraction of the Earth’s surface, many organisms living here should (and do) have special adaptations for that case, including actions of many biomolecules of very different chemical natures.

Given this great importance, it is natural that the processes of water freezing and ice melting have been investigated for a very long time. There are many excellent papers considering different aspects of these processes and the participation of biomolecules in them, but many important details of the emergence of ice—especially in living bodies—are still unclear.

In this article, we focus on the nucleation of ice rather than on its growth (because there is “no pregnancy without conception”, and waiting for the ice nucleus to appear takes much more time than its growth; see below). Hence, the “freezing temperature” refers to the “ice nucleation temperature” everywhere in this paper. We evaluate the temperature dependence of the characteristic time of nucleation of the “new” crystalline phase (i) in the volume and (ii) on surfaces adjacent to the “old” liquid phase (being interested in the vicinity of the point of thermodynamic equilibrium of these phases), and then discuss some experimentally investigated biological and non-biological agents that can drastically reduce this typically long (when temperature approaches 0 °C) time.

It is common knowledge that water freezes at 0 °C. More precisely, 0 °C is the temperature at which ice coexists with water, but this does not mean that water begins to freeze at 0 °C when cooled. Freezing is a first-order phase transition [1,2], which can be extremely slow; thus, water can exist as a supercooled liquid at subzero temperatures for a long time rather than freezing at exactly 0 °C (we did our best to find a paper on the kinetics of water freezing near 0 °C, but failed).

The formation of ordered phases from melts is described in [1,2,3,4]. For three-dimensional (3D) crystals, this is a first-order phase transition. The emergence of an ordered 2D layer is not, strictly speaking, a first-order phase transition, but the ordered layer is quite similar to a crystal [5], making it possible to consider it in the same way as the nucleation and growth of 3D crystals.

According to the classical nucleation theory, the basic estimate of the nucleation time for phases arising from a first-order phase transition can be achieved using the transition state theory [6,7] when the nucleus of a new phase is sufficiently large [8] and there are no flows or shock waves [9] in the system. For this purpose, one must trace the change in free energy occurring during the formation of a piece of the new phase.

This change for a piece of the *d*-dimensional (*d* > 1) new phase of *n* >> 1 particles (Figure 1) is known to be estimated [1,2,3,4] as follows:(1)$$G\left(n\right)\approx n∆\mu +\alpha_{d}n^{1-1/d}B_{d},$$
where ∆μ is the chemical potential of a molecule in the “new” (arising) phase minus that in the “old” one, so that ∆μ<0 when the “new” phase is more stable than the “old” one, and ∆*μ* = 0 at the point of thermodynamic equilibrium of these phases; $B_{d}>0$ is the additional free energy of one molecule on the surface (for *d* = 3) or perimeter (for *d* = 2) of the “new” phase; and $\alpha_{d}n^{1-1/d}$ is the number of molecules on the surface or perimeter of a compact piece of the new phase of *n* particles. Given d=2, $\alpha_{2}=\sqrt{4\pi }\approx 3.5$ for a circle, and $\alpha_{2}=2d=4$ for a square; with d=3, $\alpha_{3}=\sqrt[{3}]{36\pi }\approx 4.8$ for a sphere, and $\alpha_{3}=2d=6$ for a cube. Then, for further approximate estimates, we can take the average values ($\alpha_{2}\approx 3.8$ and $\alpha_{3}\approx \;$5.4). The free energy of non-compact (with larger $\alpha_{d}$) intermediates of new phases is higher (because $B_{d}>0$) than that of the compact ones; thus, when estimating the time of nucleation of new phases using the transition state theory [6,7,8], we may ignore slow paths going through the non-compact structures.

## 2. The Formation of Ice: Theory

The nucleation of a new phase is a multistep reaction (Figure 2). It begins with the sticking together of a few particles of the liquid in a configuration that allows further growth of the new phase.

We are interested in the case when the “new” phase is stable (∆μ<0) but the temperature *T* is just a little lower than $T_{0}$—the phase equilibrium point $T_{0}\approx 273\;K$ for water/ice transition. When $T-T_{0}<0$ and $\left|T-T_{0}\right|\ll T_{0}$, $0<-∆\mu \ll k_{B}T$ ($k_{B}$ being the Boltzmann constant), and $-∆\mu \ll B_{d}$.

In a multidimensional (*d* > 1) system, $G\left(n\right)$ first grows with increasing *n*, passes through its maximum—which occurs in the transition state corresponding to the Gibbs’ “critical nucleus” of the new phase—and then decreases [1,2,3,4,8]. The $G\left(n\right)$ is maximal when ${\left.\frac{dG}{dn}\right|}_{n=n_{0}}=∆\mu +\frac{d-1}{d}\alpha_{d}n_{0}^{-1/d}B_{d}=0$, i.e., when the critical nucleus includes $n_{0}\approx {\left(\frac{d-1}{d}\alpha_{d}\frac{B_{d}}{-∆\mu }\right)}^{d}$ particles, and the “seed” (which is the minimally stable—i.e., having $G\left(n\right)=0$ at n>0—piece of the “new” *d*-dimensional phase) contains
(2)$$n_{\mathrm{seed}}={\left(\alpha_{d}\frac{B_{d}}{-∆\mu }\right)}^{d}=n_{0}{\left(\frac{d}{d-1}\right)}^{d}$$
particles. Thus, the diameter of the “seed” corresponds to a row of about $n_{\min}\approx {\left(n_{\mathrm{seed}}\right)}^{1/d}=\alpha_{d}\left(\frac{B_{d}}{-∆\mu }\right)$ particles.

The free energy of the transition state at *d >* 1 and *T* close to $T_{0}$ is
(3)$$G_{\mathrm{nucl}}^{\#}\left(T\right)\equiv G\left(n_{0}\right)=\frac{\alpha_{d}B_{d}}{d}{\left(\frac{d-1}{d}\alpha_{d}\frac{B_{d}}{-∆\mu }\right)}^{d-1}=\frac{\alpha_{d}B_{d}}{d}n_{0}^{d-1}.$$

At $T\to T_{0}$, ∆μ→0, so that $G_{\mathrm{nucl}}^{\#}\to \infty$ and $n_{0}\to \infty$, $n_{\mathrm{seed}}\to \infty$; the latter allows neglecting details of the critical nucleus and seed structures [8] when *T* is close to $T_{0}$.

### 2.1. The Time Required for the Nucleation of a New Phase

Let us focus on the time required for the initiation of ice and, for reasons that will become clear soon, neglect the time of its growth for the time being.

According to the transition state theory [6,7,8], the time of appearance of an ice nucleus around a given H_2_O molecule can be estimated as $\sim \;\tau \cdot \exp\left(\frac{G_{\mathrm{nucl}}^{\#}\left(T\right)}{k_{B}T_{0}}\right)$, where τ is the time spent to cross the very top of the activating barrier. We consider τ later on; for now, it is enough to mention that the exact value of this term is of secondary importance as compared to the exponent: τ is not less than $\tau_{0}\sim 10^{-12}$ s—the typical time of thermal vibrations at 0 °C ($\tau_{0}$~$h/k_{B}T_{0}$~10^-12^ s, *h* being the Planck constant), while $\exp\left(\frac{G_{\mathrm{nucl}}^{\#}\left(T\right)}{k_{B}T_{0}}\right)$ can be infinitely large at $T\to T_{0}$.

At the melting point ($T_{0}$ ≈ 273 K at 1 atm pressure), ∆μ≡0 by definition, andat the temperature $T_{0}-∆T$ (where $0\;\leq \;∆T\ll T_{0}$), the chemical potential $∆\mu =-∆S_{\left(1\right)}\cdot \left(-∆T\right)=-∆H_{\left(1\right)}\left(\frac{-∆T}{T_{0}}\right)$ according to classical thermodynamics (where $∆S_{\left(1\right)}$ and $∆H_{\left(1\right)}$ are the entropy and enthalpy of water freezing per molecule, respectively). Experimentally, for water, the enthalpy of freezing per mol is ∆H≈−6.0 kJ/mol $\approx -2.6RT_{0}$ (and thus, $∆H_{\left(1\right)}\approx -2.6k_{B}T_{0}$ per molecule) [10], so that
(4)$$\frac{∆\mu }{k_{B}T_{0}}=\frac{∆H_{\left(1\right)}}{k_{B}T_{0}}\left(\frac{∆T}{T_{0}}\right)\;\approx \frac{-∆T}{100{}^{\circ}}$$(because 1 °K ≡ 1 °C, we do not specify—here or below—“°K” or “°C” when this is not required).

The value of τ can be approximately estimated as follows:(5)$$\tau \;\approx \;\left[\tau_{0}\cdot \;\exp\left(\frac{\varepsilon }{2k_{B}T_{0}}\right)\right]\cdot \frac{1}{\alpha_{d}n_{0}^{1-1/d}}\sqrt{2\pi }{\left\{\frac{\partial^{2}}{\partial n^{2}}{\left[\frac{-G\left(n\right)}{k_{B}T_{0}}\right]}_{n=n_{0}}\right\}}^{-1/2}\;,$$
where $\tau_{1}\equiv \tau_{0}\cdot \;\exp\left(\frac{\varepsilon }{2k_{B}T_{0}}\right)$ is the time of diffusive exclusion of the H_2_O molecule from the ice surface or its inclusion in this surface at *T* ≈ *T*_0_ [2] (ε≈51 kJ/mol [10] being the energy of ice sublimation); thus, $\tau_{1}\sim 10^{-7}s$; $\alpha_{d}n_{0}^{1-1/d}$ is the number of water molecules on the surface of the critical nucleus (which approximately equals the number of possible parallel nucleus growth pathways, since water molecules go to and from the nucleus more or less independently), and $\sqrt{2\pi }{\left\{\frac{\partial^{2}}{\partial n^{2}}{\left[\frac{-G\left(n\right)}{k_{B}T_{0}}\right]}_{n=n_{0}}\right\}}^{-1/2}$ plays the role of the Zeldovich factor [9], which takes into account the number of reaction steps spent to cross the top of the activating barrier (for mathematical details, see Ref. [11]). As a result,
(6)$$\tau \;\sim \;10^{-7}s\cdot \sqrt{\frac{2\pi }{\alpha_{d}^{3}\;\cdot \;\frac{d-1}{d^{2}}\;\cdot \;\frac{B_{d}}{k_{B}T_{0}}}}\;\cdot n_{0}^{\frac{d-3}{2d}};$$
so $\tau \sim 0.5\cdot 10^{-7}s$ at *d* = 3 and $\tau \sim 0.7\cdot 10^{-7}\cdot n_{0}^{\frac{-1}{4}}\approx 0.5\cdot 10^{-7}\sqrt{∆T/100{}^{\circ}}\;s$ at *d* = 2.

To begin with, we consider the “homogeneous” (i.e., in bulk water) ice nucleation, show that ice cannot arise in this way above ≈−35 °C, and then turn to the “heterogeneous” (occurring at the water border) ice nucleation.

### 2.2. The Homogeneous Nucleation of Ice: Ice Formation in Bulk Water

An experimental estimate of $B_{3}$ ($\approx 0.85\;k_{B}T_{0}$ near $T_{0}=273\;K$) follows from the ice–water interface free energy, ≈32 erg/cm^2^ [12], and since an H_2_O molecule occupies ≈10 Å^2^ of the interface, $B_{3}\approx \;320\times 10^{-16}\mathrm{erg}\approx 1.9\;\mathrm{kJ}/\mathrm{mol}\approx \;0.85\;k_{B}T_{0}$ per surface H_2_O molecule. Thus,
(7)$$\frac{B_{3}}{-∆\mu }=\left(\frac{B_{3}}{-∆H}\right)\left(\frac{T_{0}}{∆T}\right)\approx 0.32\left(\frac{T_{0}}{∆T}\right)\approx \frac{85{}^{\circ}}{∆T},\;$$
and the transition state (nucleus) free energy for the 3D ice crystal formation is
(8)$$\frac{G_{3D\_\mathrm{nucl}}^{\#}}{k_{B}T_{0}}\approx 2.0{\left(\frac{T_{0}}{∆T}\right)}^{2}\approx 14{\left(\frac{100{}^{\circ}}{∆T}\right)}^{2}.$$

When ∆T is small, $G_{3D\_\mathrm{nucl}}^{\#}$ is extremely high: at −5 °C (∆T=5°), $G_{3D\_\mathrm{nucl}}^{\#}\approx 13,000$ kJ/mol (which is about 50 times higher than the enthalpy of interaction of non-polymeric ions and molecules with their environment [13]); only at ∆T=35°, $G_{3D\_\mathrm{nucl}}^{\#}\approx 260$ kJ/mol, and approaches this enthalpy.

In the volume containing $N_{V}$ water molecules, a nucleus can arise around any of them, and the characteristic time of the emergence of one and only one 3D ice nucleus in this volume can be estimated as follows:(9)$$TIME_{3D}\;\sim \;\tau \cdot \exp\left(\frac{G_{3D\_\mathrm{nucl}}^{\#}}{k_{B}T}\right)/N_{V}\;\sim \frac{10^{-7}\cdot 0.5}{N_{V}}\cdot \exp\left(14{\left(\frac{100{}^{\circ}}{∆T}\right)}^{2}\right)\;s.$$

Here, we used Equations (3) and (6), neglecting insignificant details such as the change in $B_{3}$ with temperature.

The characteristic times of the homogeneous (occurring in bulk water) ice formation at different temperatures in vessels of different volumes are illustrated in Table 1.

If ice forms not around a water molecule, but around some other, “foreign” small molecule or ion that attracts ice very strongly, then its nucleation accelerates, but this may only manifest itself in observable phenomena at temperatures below −30 °C.

Concluding this section, it is worthwhile to note that the diameter of the ice “seed” corresponds (see Equations (2) and (7)) to the row of ≈50 water molecules (≈150 Å) at −10 °C, and ≈10 water molecules (≈30 Å) at −40 °C. Such seeds contain thousands of water molecules, allowing one to neglect the details of the seed structure [8].

### 2.3. The Time of Ice Growth after the Nucleation

So far, we have taken into account only the time of ice initiation, but not the time of its growth. The reason for this is that growth is a relatively fast process. For a surface H_2_O molecule, the rates of its attachment to and detachment from the ice surface are $\frac{1}{\tau_{1}}\sim 10^{7}s^{-1}$ and $\frac{1}{\tau_{1}}\exp\left(\frac{+\Delta \mu }{k_{B}T}\right)$, respectively, so that $\frac{1}{\tau_{1}}\left[1-\exp\left(\frac{+\Delta \mu }{k_{B}T}\right)\right]\approx \frac{1}{\tau_{1}}\left(\frac{-\Delta \mu }{k_{B}T}\right)$ is the resulting rate of binding of the H_2_O molecule to ice [2]. Since after the seed formation the remaining H_2_O molecules attach to ice more or less independently, $\tau_{1}\left(\frac{k_{B}T}{-\Delta \mu }\right)\approx 10^{-7}s\cdot \left(\left(\frac{T}{273^{}\;K}\right)\cdot \frac{100^{}\;K}{∆T}\right)$ is both the time of binding of one H_2_O molecule to ice and the time of ice growth by one new layer. Thus, the time of addition of one new layer of water molecules to the ice at −35 °C is ~3 × 10^−7^ s. Then, ~3 × 10^6^ new ice layers (~1 mm of ice) join the piece of ice per second, and the ice will be 30 mm thick within half a minute after the nucleation starts, while the ice initiation in a glass at −35 °C takes a year. Thus, the time of ice growth is negligible as compared to the time of its emergence.

The above shows that ice can never arise via homogeneous nucleation in bulk water [15,16,17] at temperatures above −30 °C (and according to [18], freezing of water droplets in the atmosphere only occurs below −35 °C, with the maximum rate of their freezing being observed between −42° and −46 °C). A comparison with Table 1 shows that the error in our ∆T estimates does not exceed 20%.

It is quite another matter when ice arises not in bulk water, but on its surface, via “heterogeneous” [1,2,3,4,7] nucleation. Any ice-binding surface—even that of dust particles—can drastically accelerate freezing and elevate the freezing temperature (cf. Table 1 and Table 2 below), but as we will see later, not every surface can provide water freezing at a temperature very close to 0 °C.

### 2.4. The Heterogeneous Nucleation of Ice: Ice Formation at a Surface

Let us now consider the formation of ice by heterogeneous nucleation. This occurs at the border of water, which is an ice-binding surface (otherwise, ice will not grow there; we do not consider non-ice-binding surfaces at all).

If ice’s adhesion to the surface is stronger than that of water, the ice will form a monomolecular layer there even at temperatures > 0 °C, but massive ice growth (which is our sole interest) can occur on this ice surface only at temperatures below 0 °C.

Thus, we have to consider the ice nucleation on the borders covered with ice.

The border free energy of a piece of the “new” monomolecular ice layer sticking to the “old” ice surface is solely determined by the perimeter of this new layer, because the emerging free energy of the outer (in Figure 1b and Figure 2) surface of the new layer is compensated by the disappearing free energy of the “old” ice surface, which is now covered with the new ice layer. According to Equation (3), the border free energy determined by the perimeter of the nucleus of this “new” layer is $G_{2D\_\mathrm{nucl}}^{\#}\approx 3.6\cdot B_{2}\left(\frac{B_{2}}{-∆\mu }\right)$, where *B*_2_ is the additional free energy of the perimeter’s molecules.

The value of *B*_2_ depends on interactions between molecules within the emerging ice layer. The pattern of these interactions may depend on the structure of the surface where the new layer of ice grows.

Let us start by examining the ice nucleation on a smooth, flat surface (Figure 1b), i.e., assuming that interactions within the “new” layer are the same as in a layer in the 3D body (Figure 1a).

In the case $B_{2D\_\mathrm{smooth}}$, the additional free energy of a molecule on the perimeter of the layer covering a smooth, flat surface is close to $B_{3}$—the additional free energy of a molecule at the border of a 3D body (Figure 1a)—so that $B_{2D\_\mathrm{smooth}}\approx 0.85\;k_{B}T_{0}$. Notably, due to the attraction of the edge molecules of the new layer to the underlying surface, $B_{2D\_\mathrm{smooth}}$ may be slightly lower than $B_{3}$ [15].

Using Equations (2) and (7), we can estimate the diameter of ice “seeds” formed on smooth, flat surfaces; they correspond to the rows of ≈80 water molecules (≈240 Å) at −2 °C, ≈40 water molecules (≈120 Å) at −4 °C, ≈20 water molecules (≈60 Å) at −8 °C and ≈10 water molecules (≈30 Å) at −16 °C. Such seeds contain thousands of water molecules, allowing us to neglect details of the seed structure [8].

The free energy of the ice nucleus on a smooth, flat surface can be estimated using Equations (3) and (7):(10)$$\frac{G_{2D\_\mathrm{nucl}\_\mathrm{smooth}}^{\#}}{k_{B}T_{0}}\approx 0.95\left(\frac{T_{0}}{∆T}\right)\approx \frac{260{}^{\circ}}{∆T}.$$

When ∆T is small, $G_{2D\_\mathrm{nucl}\_\mathrm{smooth}}^{\#}$ is high: at ∆T=0.2°, $G_{2D\_\mathrm{nucl}\_\mathrm{smooth}}^{\#}\;\approx 3000$ kJ/mol (which is 10 times higher than the enthalpy of the interaction of non-polymeric ions and molecules with their environment [13]), and $G_{2D\_\mathrm{nucl}\_\mathrm{smooth}}^{\#}$ only approaches this enthalpy at ∆T≈2°.

Thus, the characteristic time of appearance of the first 2D ice nucleus on a smooth surface covered with $N_{S}$ water molecules can be estimated as follows:(11)$$TIME_{2D\_\mathrm{smooth}}\;\sim \;\tau \cdot \exp\left(\frac{G_{2D\_\mathrm{nucl}\_\mathrm{smooth}}^{\#}}{k_{B}T}\right)/N_{S}\;\sim \frac{\;0.5\cdot 10^{-7}\sqrt{∆T/100{}^{\circ}}}{N_{S}}\cdot \exp\left(\frac{260{}^{\circ}}{∆T}\right)\;s.$$

Because *N*_S_—the number of aqueous molecules on the surface—is not unequivocally determined by *N*_V_—the number of molecules in the volume—it makes sense to consider two reasonable limits: *N*_S___min_ ≈ (*N*_V_)^2/3^ (the volume cross-section) and *N*_S_max_ ~ *N*_V_/1000 (assuming that the wall of the vessel has ice-binding protrusions, that many water molecules contact ice-binding dust floating around, etc.).

The characteristic times for heterogeneous ice nucleation on flat and smooth surfaces of various sizes are presented in Table 2 for different temperatures.

We can see that (i) the time of the heterogeneous ice nucleation on smooth, flat surfaces is temperature-dependent, although not as much as that of the homogeneous nucleation, and also tends towards infinity when the temperature approaches 0 °C; (ii) at subzero temperatures from 0 °C to −2.5 °C, ice cannot arise in a reasonable time via heterogeneous nucleation on smooth ice-binding surfaces, even in a large lake full of ice-binding dust; in microscopic vessels such as animal and plant cells, ice cannot arise at all on smooth, flat, ice-binding surfaces at temperatures above −4.5 °C.

Our experiments on water freezing in 15 plastic test tubes [16,17] showed that in 0.5 mL of distilled water, ice arises at temperatures from −9.0 °C to −14.7 °C. Each set (test tube + sample) had its own ice-nucleating temperature, which was well reproducible if several freezing–thawing cycles were carried out with the same sets, while different sets had somewhat different ice-nucleating temperatures.

The observed temperatures (−9.0 °C to −14.7 °C) are incompatible with “in bulk” nucleation (Table 1), and are compatible with freezing on surfaces only if their ice-binding parts cover a very small portion of the water borders (nesting only ~300—300,000 water molecules). It should be noted that these temperatures obtained by our team are higher than those usually reported for ice nucleation of water in plastic tubes (e.g., <−15 °C [19] or <−18.5 °C [20]). This may be because the authors of [19,20] tried to eliminate the greatest possible amount of potential ice-nucleating dust microparticles from water by autoclaving at +121 °C, avoiding contamination, etc. We did not try to do this, because the ice-binding protein does not inhibit freezing of the buffer itself (with its possible ice-nucleating dust microparticles), but inhibits the action of added ice-nucleating agents of different origins (see Table 3 below).

The above theory and experiments demonstrate that freezing at 0 °C (or very close to 0 °C) cannot be explained by the presence of smooth, flat, ice-binding surfaces, and requires ice-nucleating surfaces of some special shape, as considered below.

It should be added that our other experiments using the same test tubes showed that ice melts exactly at 0 °C [16,17], without any lag caused by the first-order phase transition (which is observed for freezing). The absence of the lag in melting is a well-known effect [1,9,15,21]. This happens because the free energy of the ice–air border (109 erg/cm^2^) is equal to the sum of free energies of the ice–water (32–33 erg/cm^2^) and water–air (76 erg/cm^2^) borders [9,15] (the numerical data are taken from [10,12,22,23]); therefore, the surface of ice is similar to supercooled liquid water [21].

## 3. The Ice-Nucleating Surfaces

### 3.1. The Simplest Mechanistic Model

To provoke the ice nucleation at nearly 0 °C, the underlying surface must disrupt the interactions between molecules in the arising ice layer and replace them with interactions with neighboring ice layers. The weakening of interactions within the arising layer can decrease or even nullify the resulting *B*_2_, as well as the $G_{2D\_\mathrm{nucl}}^{\#}$ values (see Equation (3)).

In the simplest and very schematic mechanistic model shown in Figure 3, the disruption of the interactions between molecules in the arising ice layer is achieved when the underlying surface is corrugated (see Figure 3a) or fluted (see Figure 3b) [15]. Then, molecules of the arising layer have strong interactions with the underlying surface, but no or almost no interactions between themselves (i.e., within the emerging layer) in one (“corrugated surface”, Figure 3a) or even in both (“fluted surface”, Figure 3b) directions.

In the latter case (Figure 3b), the resulting $B_{2\_\mathrm{fluted}}$ value is very small:(12)$$B_{2\_\mathrm{fluted}}\ll B_{3}$$
because of weak contacts between molecules of the arising layer (Figure 3c).

In the former case (Figure 3a), $B_{2\_\mathrm{along}\_\mathrm{groove}}\approx B_{3}$, while $B_{2\_\mathrm{across}\_\mathrm{grooves}}\sim B_{2\_\mathrm{fluted}}\ll B_{3}$, and the effective additional free energy of the molecule at the perimeter of ice
(13)$$B_{2\_\mathrm{corrug}}=\sqrt{B_{2\_\mathrm{along}\_\mathrm{groove}}\cdot B_{2\_\mathrm{across}\_\mathrm{grooves}}}$$
is also small.

If $B_{2\_\mathrm{fluted}}$ or $B_{2\_\mathrm{corrug}}$ is, for example, 10 times lower than $B_{3}$, the time of ice nucleation on such a surface is close to that given in Table 2—but only when the temperatures are 100 times closer to 0 °C (see Equations (3) and (11)) than to −3°—−5 °C to given in Table 2, and the freezing occurs at −0.03°—−0.05 °C.

### 3.2. The Atomic Structure of Ice Has Different Ways of Attachment to Different Ice-Binding Surfaces

This is illustrated in Figure 4 and Figure 5. It can clearly be seen that one way of attachment contains hydrogen bonds (the strongest interactions in ice) only between the neighboring quasi-layers of water molecules, but not within these quasi-layers (see Figure 4), while another way of attachment contains hydrogen bonds both between and within the neighboring quasi-layers of water molecules (see Figure 5).

This difference should lead to different additional free energies of the perimeters of different quasi-layers and, therefore, to different rates of ice nucleation at different ice-binding surfaces.

Numerical estimates of these effects are beyond the scope of this paper, but it should be noted that recent experimental studies (which did not concern near-zero temperatures, however) indicate that a surface with an atomic-scale corrugation strongly favors ice nucleation [25], although the size of ice-nucleating sites should be hundreds of times greater than that of an atom (see the text above Equation (10)).

Without the ice-nucleating surfaces (and without in-water shock waves; see [9]), water cannot freeze in a closed vessel when the ambient temperature is very close to 0 °C. However, it should be noted that in open vessels such as puddles, the water-to-air evaporation at an air temperature of 0 °C can decrease the water surface temperature down to the dew point [26,27], which is below −5 °C at a relative air humidity below ≈75% [15,26,28], allowing ice nuclei to arise on ice-binding surfaces (e.g., on specks of floating dust) at an ambient temperature of 0 °C.

## 4. A Brief Overview of Experiments on Ice Nucleation at High Subzero Temperatures

The ability of some surfaces to bind and nucleate ice has long attracted human attention. According to Faraday [29], this property is possessed by flannel, but not by gold. In 1947, Bernard Vonnegut (a brother of the writer Kurt Vonnegut, who authored the famous “*Cat’s Cradle*” [30]) discovered [31] the ice-nucleating effect of silver iodide (AgI), which is now considered to be one of the most powerful inorganic nucleators of ice (acting up to −3–−4 °C [32]). Then followed less powerful but much more abundant inorganic ice nucleators such as metals [25] and metal oxides [33], along with some organic substances [34,35,36] and biological objects such as pollen and bacteria [37,38,39,40]. For our experiments, we chose two ice nucleators that are very different in chemical nature: the inorganic compound CuO (in the form of powder), and the bacterium *Pseudomonas syringae*, which now seems to be the most powerful nucleator of ice [38,39,41,42].

Unlike nucleators facilitating ice formation, the much better-known antifreeze proteins (AFPs) hinder the appearance and/or growth of ice, or inhibit ice recrystallization [43,44]; at least some of them can block the action of ice nucleators [45]. These ice-binding proteins (IBPs) are synthesized by various organisms that survive at subzero temperatures. They were first found in the blood of fish living in polar waters [46], and then in many other organisms, from bacteria [47] to fungi [48], plants [49], insects [50], etc. The mechanism of their action remains rather obscure [44].

Our experiments were performed using a mutant form (mIBP83) of the ice-binding protein cfAFP described in [51,52]; cfAFP is an antifreeze protein of the spruce budworm (*Choristoneura fumiferana*)—a moth whose larvae winter in needles [53] at a temperature of about −30 °C. The cfAFP mutant, mIBP83 [54,55], was used because along with retaining the ability of ice binding, it is less susceptible to aggregation during isolation and purification than WT cfAFP, thus fitting the experiments better.

To investigate the nucleation of ice under the action of different surfaces, we measured ice-nucleating temperatures for a buffer per se and with added ice nucleators (either CuO or *P. syringae*), and then we tested the influence of the mIBP83 ice-binding protein on this temperature. In all cases, the samples (0.5 and 1.0 mL) were placed into standard plastic (polypropylene) test tubes; thus, the walls of the test tubes as well as some particles of dust could also participate in the process of ice nucleation.

The experimental results are presented in Table 3. One can see that the buffer per se freezes between −8 and −10 °C, and that the addition of either CuO or *P. syringae* to the buffer results in an increase in the freezing temperature (to between −1 and −2 °C in the case of *P. syringae*, and to between −5 and −7 °C in the case of CuO). It should be noted that, in spite of the fact that CuO nucleates ice at lower temperatures than the conventional AgI (for this comparison, the corresponding literature data for AgI are also included in Table 3), the ice-nucleating activity of CuO observed in our experiments was higher than that of many other inorganic ice nucleators described in the literature [33,56,57].

**Table 3 biomolecules-12-00981-t003:** Freezing (ice-nucleating) temperatures for explored liquids in standard plastic (polypropylene) test tubes.

Basic Liquid	Added Nucleator	Ice-Binding (Antifreeze) Protein	Freezing (Ice-Nucleating) Temperature	Reference
Sodium phosphate buffer 20 mM pH 7, volumes 0.5 and 1.0 mL	-	-	−8–−10 °C	Our experiment
	-	mIBP83	−8–−9 °C	Our experiment
	AgI	-	−3–−3.5 °C	[31,32]
	AgI	No experiment	No experiment	-
	CuO	-	−5–−7 °C	Our experiment
	CuO	mIBP83	−9–−11 °C	Our experiment
	*P. syringae*	-	−1–−2 °C	[41], our experiment
	*P. syringae*	mIBP83	−4–−7 °C	Our experiment

The main result of the experiments with mIBP83 ice-binding protein was that the ice-binding protein did not inhibit freezing of the buffer itself (with its possible ice-nucleating dust microparticles), but inhibited the action of added ice-nucleating agents of different origins (see Table 3). In other words, it is obvious that the tested ice nucleators eliminate the significant preliminary overcooling of the freezing liquid, and the antifreeze protein hinders the action of these ice nucleators.

## 5. Materials and Methods

We investigated the freezing of water (more precisely, 20 mM sodium phosphate buffer, pH 7), including freezing in the presence of the nucleators CuO and *P. syringae*, using the setup and method of experiments described in detail in [41] and [58].

Experiments on freezing and melting were carried out using a Julabo F-25 thermostat (Germany). The thermostat and measuring thermometers (thermocouples) were checked using an LT-300-N TERMEX laboratory thermometer (Russia), at a resolution of 0.01 °C, and with an accuracy of ±0.05 °C.

Freezing of the solutions was carried out in standard plastic (polypropylene) microcentrifuge test tubes (1.7 mL, Cat. No. 3621, Costar^®^). The liquid volume was 0.5 and 1 mL.

Copper(II) oxide (CuO) was from Reachem (Moscow, Russia). It was added to the test tubes in amounts of 2 mg. This non-soluble CuO powder was at the bottom of the test tubes.

*P. syringae* cells (*Pseudomonas syringae* pv. *syringae*) were grown on medium L (yeast extract 5.0 g/L; peptone 15.0 g/L; NaCl 5.0 g/L) at a temperature of +26 °C. Cells were grown in a liquid medium up to the cell density of 1.0 optical units (by absorption at 600 nm), and then precipitated on a centrifuge at 6000 g, and washed twice with a solution of 20 mM Tris-HCl (pH 7.5). The initial cell solution was diluted with a buffer of the same composition to the desired cell density (0.1 optical units). The concentration of *P. syringae* cells in the experiments was controlled by absorption at 600 nm.

The experiments were repeated at least 10 times, which made it possible to determine the ice-nucleating temperature range for the used solutions.

## 6. Conclusions

We report that the kinetics of ice nucleation (i) prevent ice formation in bulk water at temperatures above −35 °C, (ii) prevent ice emergence on smooth ice-binding but not ice-nucleating surfaces at temperatures above −4 °C, and (iii) allow ice nucleation and growth on ice-binding surfaces with a grooved (corrugated or fluted) atomic structure at nearly 0 °C.

In the absence of in-water shock waves or water overcooling by evaporation from an open vessel, the presence of an ice-binding surface with a specific (fluted or corrugated) atomic structure that disrupts interactions between molecules within the ice layers arising on this surface seems to be the only possibility to achieve ice nucleation at or very close to 0 °C in any closed vessel, including animal or plant fluids, tissues, and cells.

Our experiments show that the ice nucleation temperature is about −10 °C for a buffer in plastic test tubes, while with CuO added, it ranges from −5 to −7 °C (thus, CuO appears to be a powerful inorganic ice nucleator), and in the presence of the ice-nucleating bacterium *Pseudomonas syringae* the ice nucleation temperature is about −1 to −2 °C. The addition of an antifreeze protein inhibits the action of both of these ice-initiating agents.

## Figures and Tables

**Figure 1 biomolecules-12-00981-f001:**
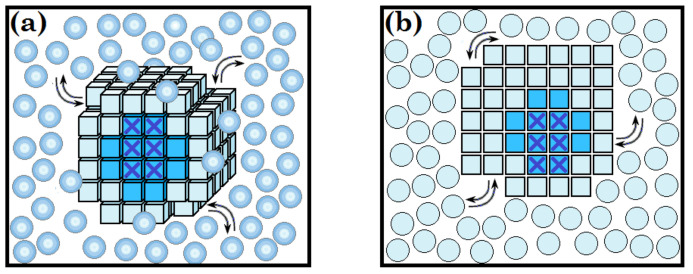
A schematic presentation of a compact structured piece of the emerging 3D phase in bulk liquid (**a**), and of the emerging 2D phase at a surface (**b**): The pieces of the “new” structured phases shown as cubes in panel (**a**) and squares in panel (**b**) arise from “free” particles of the liquid (balls in panel (**a**) and circles in panel (**b**)). Dark blue cubes and squares represent the “seeds” in the structured pieces—that is, the smallest minimally stable (as compared to the liquid) parts of the structured phases; light blue cubes and squares are particles that stick to these “seeds” later. Dark blue crosses are the “nuclei” within the “seeds”, i.e., entities displaying the minimal stability during the structuring process.

**Figure 2 biomolecules-12-00981-f002:**
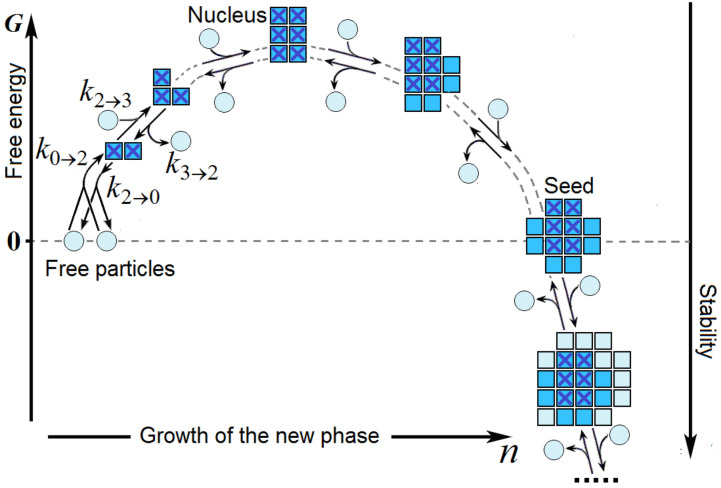
A sketch of the crystal formation from free particles: The reaction is initiated by the sticking together of a few particles in a configuration that allows further growth of the ordered phase. Rather than the initiation, the new phase appearance rate is determined by the formation of a “nucleus” of this phase (shown by crosses) that corresponds to the activation barrier, i.e., has the maximal free energy *G* and, thus, the minimal stability during this process. *k*_0__→2_, *k*_2__→3_,… are the rate constants of the initiation and further growth of the new phase; …, *k*_3__→2_, *k*_2__→0_ are the rate constants of the new phase’s decrease and subsequent disappearance (see text). For other designations, see Figure 1.

**Figure 3 biomolecules-12-00981-f003:**
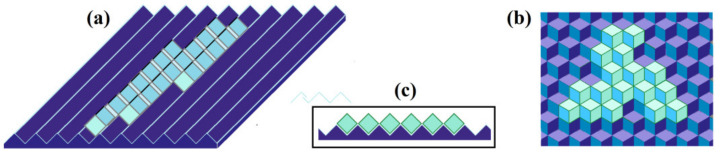
A scheme of the arising ice layers (light blue) on a corrugated (**a**) or fluted (**b**) ice-binding surface (blue–violet). The cross-section (**c**) illustrates the weakness of molecular contacts within the layers covering such surfaces. Due to this weakness, the additional border free energy at the perimeter of the arising ice layers is low; therefore, these new layers are not as compact as that in Figure 1b.

**Figure 4 biomolecules-12-00981-f004:**
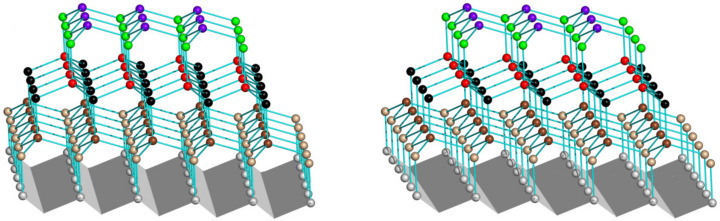
A stereo drawing of a piece of “normal” hexagonal ice (ice I*h*, [24]) on a corrugated underlying surface (highlighted in gray) fitted to basal planes of this piece, which are horizontal in this figure. Oxygen atoms are shown as balls. In different quasi-layers they are shown in different colors. Hydrogen atoms are not shown. Lines between O atoms correspond to O-H-O hydrogen bonds.

**Figure 5 biomolecules-12-00981-f005:**
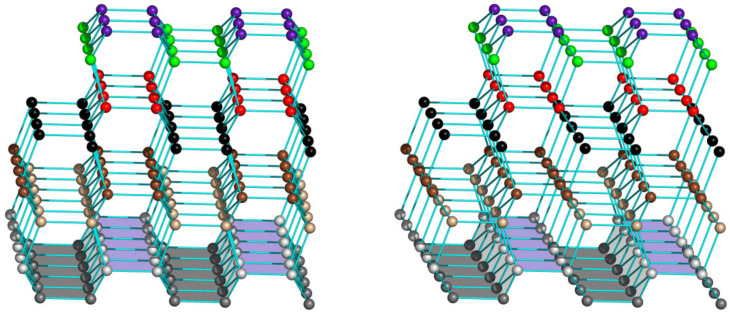
A stereo drawing of a piece of the same “normal” hexagonal ice (ice I*h*, [24]) on another underlying surface (highlighted in gray and violet–gray). This underlying surface is fitted to prism planes of this piece of ice, which are horizontal in this figure. Atoms and hydrogen bonds are shown as in Figure 4.

**Table 1 biomolecules-12-00981-t001:** Time of homogeneous ice nucleation in bulk water in the absence of ice-binding surfaces ^†^.

Volume (V)	Water Molecules in the Volume, *N*_V_	$TIME_{3D}\left(∆T\right)$		
		**−35** **°C** **:** **∆** * **T** * **= 35°**	**−40****°C****:** **∆*****T*** = **40****°**	**−50****°C****:** * **∆*****T*** = **50****°**
A lake 30,000 m^3^	10^33^	100 years	0.01 s	10^−16^ s
A glass 30 cm^3^	10^24^	10^11^ years	1 year	10^−6^ s
An eukaryotic cell or a small water droplet [14] (≈30 μm)^3^	10^15^	10^20^ years	10^9^ years	10 min

Footnotes: ^†^ The estimates are for $B_{3}$ = $0.85\;k_{B}T_{0}$ (determined for ice I*h*, which is thermodynamically stable near $T_{0}=273\;$K, although ice I*c* forms faster) [14]; however, according to data in Figure 7 of [14] and Equations (3) and (9), this can correspond to a small change in $B_{3}$—from $0.85\;k_{B}T_{0}$ at $T_{0}=273\;$K to $0.65-0.68\;k_{B}T_{0}$ at $233^{}-238\;K$. Nevertheless, no freezing at all is expected even for ice I*c* at temperatures above ≈−30 °C. * “… at −50 spittle crackled on the snow …” (Jack London. “*To Build a Fire*”).

**Table 2 biomolecules-12-00981-t002:** Time of heterogeneous ice nucleation on flat and smooth ice-binding surfaces.

Volume (V)	Number of Water Molecules		$TIME_{2D\_\mathrm{flat}}\left(∆T\right)$	
	In the Volume, *N*_V_	On the Surface, *N*_S_	**−3** **°C** ^ **†** ^ **:** **∆** * **T** * **= 3 °**	**−5****°C**^**†**^: **∆*****T*** = **5** **°**
A lake 30,000 m^3^	10^33^	*N*_S_max_ = 10^30^ *N*_S___min_ ≈ 10^22^	0.3 s 1 year	3 × 10^−16^ s * 3 × 10^−8^ s
A glass 30 cm^3^	10^24^	*N*_S_max_ = 10^21^ *N*_S___min_ ≈ 10^16^	10 years 10^6^ years	3 × 10^−7^ s * 0.03 s
A test tube 1 cm^3^	3⋅10^22^	*N*_S_max_ = 3 × 10^19^ *N*_S___min_ ≈ 10^15^	300 years 10^7^ years	3 × 10^−5^ s * 1 s
An eukaryotic cell (30 μm)^3^	10^15^	*N*_S_max_ = 10^12^ *N*_S___min_ ≈ 10^10^	10^10^ years 10^12^ years	5 min 500 min

Footnotes: **^†^** The data are tabulated under the assumption that $B_{2}=B_{3}=0.85\;k_{B}T_{0}$, although $B_{2}$ can be a little lower than $B_{3}$ (see text), and then the times and/or ∆*T* values given in the Table are slightly overestimated. Incorporation of ice-attracting non-aqueous molecules or ions on the surfaces may additionally decrease the ∆*T* values given in the table by ≈1^o^. * i.e., 1 nucleation per second in the surface layer containing ~3 × 10^14^ waters (area: ~3 cm^2^).

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
