# Peer review of "How Can Ice Emerge at 0 °C?"

_biomolecules, 2022, doi:10.3390/biom12070981_

Round 1
Reviewer 1 Report
Two mutually exclusive general comments:
If the paper is intended as a kind of introductory overview/review for the special volume, I would miss the overlap with biomolecules and with the topic of the journal "Biomolecules", namely "structures and functions of bioactive and biogenic substances, molecular mechanisms with biological and medical implications as well as biomaterials and their applications". The only biomolecules are an AFP and (presumably) the surface of a bacterium - both are not defined, and any details are hidden in citations.
If the paper is not intended as introductory, but as one contribution in the special issue, it is acceptable for the theoretical part. The experimental part adds a few experiments only to give the work some "bio" touch.
------
The paper is principally very well written, but there are several major problems that require very careful corrections.
The theoretical part is very well set up, and combines thermodynamics with kinetics without the often seen confusions, and by keeping explanations and symbols very clear. The authors make a major effort to assign real values to the constants, which makes the paper very useful also outside the field of crystallization and phase transition theory.
On top of page 2, the authors assume alpha2=3.8 and alpha3=5.4. Which geometry dooes this refer to , and can this be justified?
Above equation (4), the text states entropy and enthalpy per molecules, while it is in fact per mol, as indeed used just afterwards.
Equation (5) provides an estimate of tau, which is central to all the following conclusions. Tau and hence tau1 cannot be estimated very well. I suggest to stress this point more than just by using various "approx" symbols.
In section 3.2 the authors correctly argue with lattice-matching surfaces. On the atomic scale, this is - arguably - the only really good model we have. But on the nano- and microscale, this is apparently not a requirement for nucleation, and it does not fit to nucleation at biomolecules either. This point is unclear, although the authos make an effort at the end of the section by citing [23].
Severe problems:
The experiments with CuO add only two experimental values, all other values are from literature, and again mainly from self-citations! Is the only avaiilable value for AgI really from 1947? I see very little novel results, indeed they are for the most part from already published works from the Melnik group, [31] and [46] and [47]. Frankly, I see no point in incuding this here. Either omit all experiments, or provide a well-researched table 1 with more and updated results, not only based on own work.
Ice nucleation in plastic tubes can easily be driven to -20°C and lower, see e.g. work by C. Marcolli and coworkers from ETH Zürich, where this is standard. It is not fair to cite in this respect only the authors' work [16] and [17]. I expect more recent literature from other groups to make this point.
In section 4, Pseudomonas syringae is introduced with a citation from 1974. In the meantime, extracts from the bacterium are not tested, but actually used for the prodcution of artificial snow. Just as information, the commercial product is called "snomax".
The use of degree Celsius for absolute temperatures is misleading. It somehow suggests that one can insert the numerical value, while the formulae of course require K(elvin). In the case of temperature differences, the convention is clearly Kelvin, not Celsius. This should be corrected in the formulae. I agree with the authors that absolute temperatures in the text are very nicely readable and interpretable when given in °C, and I hope this is OK with MDPI Biomolecules.
Though I enjoyed finding citation [27], it is not related in any way to the research. In view of the habit to very heavy self-citation, and thus to suppress other authors' results, I find it slightly offensive to cite the non-science work of a brother of a scientist. Maybe one could leave [27] in, but add more relevant work from other scientists.
Some additional notes
The °C (degree Celsius) symbol is very often missing/replaced by ° (degree) in the text in and the formulae, which is probably some programming mistake.
There are various typos (e.g. affiliation 2) and incorrect or misleading expressions (e.g. the meaning of "even" in line 2 of the abstract , where one would expect the opposite , e.g. "especially", or in the intro "freezing is the ...transition", where one would expect "freezing is a ... transition". This continues throughout the paper, but without interfering with the scientific content.
Please replace [25] by a proper reference (though I personally agree that wikipedia is often very reliable!).
Author Response
Thank you very much for attentive reading and very useful comments and suggestions!
1)
Two mutually exclusive general comments:
If the paper is intended as a kind of introductory overview/review for the special volume, I would miss the overlap with biomolecules and with the topic of the journal "Biomolecules", namely "structures and functions of bioactive and biogenic substances, molecular mechanisms with biological and medical implications as well as biomaterials and their applications". The only biomolecules are an AFP and (presumably) the surface of a bacterium - both are not defined, and any details are hidden in citations.
If the paper is not intended as introductory, but as one contribution in the special issue, it is acceptable for the theoretical part. The experimental part adds a few experiments only to give the work some "bio" touch.
Response 1: This work is not a review but an original paper (contribution to the special issue "Molecular Phase Transitions in Physiology and Pathology: Freezing and Precipitation in Biology"). The experimental part is indeed smaller than the theoretical one, but we present some valuable original results in the experimental part. At first, we show that the ice-binding protein mIBP83 inhibits two ice-nucleators of different nature (an inorganic compound CuO and an ice-nucleating bacterium Pseudomonas syringae).
Moreover, it is shown in this work that CuO appears to be a powerful ice-nucleator: it nucleates ice already at minus 5oC (thus, except for AgI, almost all known inorganic ice-nucleators are weaker than CuO). Now, we stress this in the paper (before Table 3).
2)
The paper is principally very well written, but there are several major problems that require very careful corrections.
The theoretical part is very well set up, and combines thermodynamics with kinetics without the often seen confusions, and by keeping explanations and symbols very clear. The authors make a major effort to assign real values to the constants, which makes the paper very useful also outside the field of crystallization and phase transition theory.
On top of page 2, the authors assume alpha2=3.8 and alpha3=5.4. Which geometry does this refer to , and can this be justified?
Response 2: Alpha2=3.8 and alpha3=5.4 are taken simply as the average values: for alpha2, between those for a circle (=3.5) and a square (=4); for alpha3, between those for a sphere (=4.8) and a cube (=6). Now, we write that in the text (after Equation (1)) explicitly.
3)
Above equation (4), the text states entropy and enthalpy per molecules, while it is in fact per mol, as indeed used just afterwards.
Response 3: Corrected, thank you!
4)
Equation (5) provides an estimate of tau, which is central to all the following conclusions. Tau and hence tau1 cannot be estimated very well. I suggest to stress this point more than just by using various "approx" symbols.
Response 4: Thank you. We now add a word “approximately” before Equation (5).
5)
In section 3.2 the authors correctly argue with lattice-matching surfaces. On the atomic scale, this is - arguably - the only really good model we have. But on the nano- and microscale, this is apparently not a requirement for nucleation, and it does not fit to nucleation at biomolecules either. This point is unclear, although the authos make an effort at the end of the section by citing [23].
Response 5: Yes, to discuss the requirements at the atomic scale vs. size of the ice-nucleating site, we added there (to the section 3.2) the following text: “...a surface with an atomic-scale corrugation strongly favors ice nucleation, although the whole size of ice-nucleating site should be hundreds times greater than an atom, see the text above Eq. (10)”.
6)
Severe problems:
The experiments with CuO add only two experimental values, all other values are from literature, and again mainly from self-citations! Is the only avaiilable value for AgI really from 1947? I see very little novel results, indeed they are for the most part from already published works from the Melnik group, [31] and [46] and [47]. Frankly, I see no point in incuding this here. Either omit all experiments, or provide a well-researched table 1 with more and updated results, not only based on own work.
Response 6: Our experiments (and the corresponding Table 3) demonstrate an influence of the ice-binding protein mIBP83 on nucleation temperatures in buffer (in test tubes) and under addition of two selected ice nucleators of very different chemical nature (CuO and a bacterium). Now, we write (in the first paragraph of Section 4) the following; “For our experiments, we chose two ice nucleators that are very different in chemical nature: an inorganic compound CuO (in the form of powder) and a bacterium Pseudomonas syringae, which now seems to be the most powerful nucleator of ice”. Moreover, as we mentioned in Response 1, we demonstrate a good ice-nucleating effect of CuO. AgI is mentioned by us only for a comparison with our results, because AgI is a conventional, very powerful ice nucleator); we now added a more recent reference (Marcolli et al., 2016) on AgI. Now, we write all this (including purpose of the experimental part of the work and a more detailed description of the obtained results) explicitly in Sections 4–6.
7)
Ice nucleation in plastic tubes can easily be driven to -20°C and lower, see e.g. work by C. Marcolli and coworkers from ETH Zürich, where this is standard. It is not fair to cite in this respect only the authors' work [16] and [17]. I expect more recent literature from other groups to make this point.
Response 7: Yes, ice-nucleation in plastic tubes with specially purified water gives lower temperatures than those obtained by us. Now, we have added the corresponding note (in section 2.4, 3-rd paragraph from the end) with a comparison with literature data. The works cited there are not by C. Marcolli and coworkers, but we now cite C. Marcolli and coworkers’ review on AgI in the text (the first paragraph of Section 4) and in Table 3.
8)
In section 4, Pseudomonas syringae is introduced with a citation from 1974. In the meantime, extracts from the bacterium are not tested, but actually used for the prodcution of artificial snow. Just as information, the commercial product is called "snomax".
Response 8: Thank you, now we added some newer references (in the newest reference, Snomax is also considered) there, in section 4.
9)
The use of degree Celsius for absolute temperatures is misleading. It somehow suggests that one can insert the numerical value, while the formulae of course require K(elvin). In the case of temperature differences, the convention is clearly Kelvin, not Celsius. This should be corrected in the formulae. I agree with the authors that absolute temperatures in the text are very nicely readable and interpretable when given in °C, and I hope this is OK with MDPI Biomolecules.
Response 9: In formulae, there are the temperature differences, and because 1оК=1оC, we do not specify there "оK" or "оC", when this is not required. Now we write this explicitly below Eq. (4). Other issues with temperatures we now corrected, thank you!
10)
Though I enjoyed finding citation [27], it is not related in any way to the research. In view of the habit to very heavy self-citation, and thus to suppress other authors' results, I find it slightly offensive to cite the non-science work of a brother of a scientist. Maybe one could leave [27] in, but add more relevant work from other scientists.
Response 10: Now, we added more works from other scientists there and in some other places too (totally, 12 references are now added).
11)
Some additional notes
The °C (degree Celsius) symbol is very often missing/replaced by ° (degree) in the text in and the formulae, which is probably some programming mistake.
Response 11: We only use ° for temperature differences, because 1оК=1оC.
12)
There are various typos (e.g. affiliation 2) and incorrect or misleading expressions (e.g. the meaning of "even" in line 2 of the abstract , where one would expect the opposite , e.g. "especially", or in the intro "freezing is the ...transition", where one would expect "freezing is a ... transition". This continues throughout the paper, but without interfering with the scientific content.
Response 12: Corrected, thank you for careful reading!
13)
Please replace [25] by a proper reference (though I personally agree that wikipedia is often very reliable!).
Response 13: We now added a more traditional reference.
Kind regards,
Prof. Alexei V. Finkelstein,
Corresponding author
afinkel@vega.protres.ru
Reviewer 2 Report
The introduction is very short and should be extended.
The paper is presented of very confused way. Is it a review or original paper?
In the abstract the authors are speaking about experimental results. But in the paper, there is no description of the experimental procedure and experimental results are taken from previously published work.
If it is a review paper it is to be attributed as review paper and bibliometric study is to be performed.
If it is an experimental work, material and methods are to be presented.
Author Response
We thank the reviewer for his/her helpful and friendly comments.
1)
The introduction is very short and should be extended.
Response 1: Now, we have extended the Introduction.
2)
The paper is presented of very confused way. Is it a review or original paper?
In the abstract the authors are speaking about experimental results. But in the paper, there is no description of the experimental procedure and experimental results are taken from previously published work.
If it is a review paper it is to be attributed as review paper and bibliometric study is to be performed.
If it is an experimental work, material and methods are to be presented.
Response 2: Yes, it is not a review paper but an original paper. Now we have added section on materials and methods explicitly. As for our experimental results: (1) experiments with CuO are original and were not published previously. Moreover, it is shown in this work that CuO is a powerful ice-nucleator (in contrast to most other inorganic ice-nucleators except for AgI) since it “works” (nucleates ice) up to minus 5oC; (2) we show an inhibition by mIBP83 ice-binding protein of two ice-nucleators of different nature (an inorganic compound CuO and an ice-nucleating bacterium Pseudomonas syringae). Now, we explicitly stress both these results in the manuscript.
Best regards,
Prof. Alexei V. Finkelstein,
Corresponding author
afinkel@vega.protres.ru
Round 2
Reviewer 2 Report
Accept